# Analyses of L-Type Corner Joints Connected with Auxetic Dowels for Case Furniture

**DOI:** 10.3390/ma16134547

**Published:** 2023-06-23

**Authors:** Ali Kasal, Jerzy Smardzewski, Tolga Kuşkun, Ersan Güray

**Affiliations:** 1Department of Woodworking Industrial Engineering, Faculty of Technology, Muğla Sıtkı Koçman University, 48000 Muğla, Turkey; alikasal@mu.edu.tr (A.K.); tolgakuskun@mu.edu.tr (T.K.); 2Department of Furniture Design, Faculty of Forestry and Wood Technology, Poznan University of Life Sciences, Wojska Polskiego 28, 60-637 Poznan, Poland; 3Department of Civil Engineering, Faculty of Engineering, Muğla Sıtkı Koçman University, 48000 Muğla, Turkey; ersan.guray@mu.edu.tr

**Keywords:** auxetic, dowel, corner joints, furniture, FEM

## Abstract

Tests were carried out to develop and manufacture various types of auxetic dowels using 3D printing technology. These dowels were then used to connect L-type corner joint specimens for case furniture, and their strength and stiffness were analyzed through experimental, theoretical, and numerical means. In the scope of the study, eight different types of auxetic dowels including two inclusion types, two inclusion sizes, and two dowel hole diameters, as well as a reference non-auxetic dowel, were designed. Accordingly, a total of 180 specimens that included 10 replications for each group were tested; 90 were tested under tension and the remaining 90 were tested under compression. The results demonstrated that the assembly force required for the corner joints connected with auxetic dowels was significantly lower compared to non-auxetic dowels. Furthermore, the numerical and theoretical analyses yielded similar outcomes in this study. Both analyses revealed that the dowels used to connect the corner joints experienced substantial stresses during mounting and bending, ultimately leading to their failure. Upon concluding the test results, it was observed that the corner joints connected with dowels featuring rectangular inclusions exhibited superior performance when compared to those with triangular inclusions. In light of these findings, it can be concluded that further enhancements are necessary for auxetic dowels with rectangular inclusions before they can be utilized as alternative fasteners for traditional dowels.

## 1. Introduction

Furniture design must meet four essential criteria: aesthetic appeal, functionality, durability, and feasibility [1,2]. Among the various types of joints used in furniture, corner joints are the most common, connecting vertical and horizontal elements. Recently, there has been a growing interest in ready-to-assemble (RTA) furniture joints, including techniques such as welding [3,4,5], the use of auxetic materials and structures [6,7], or shrink-fitting methods [8,9,10].

Numerous studies have investigated the strength and stiffness of L-type corner joints using glued dowels or mechanical fasteners. For instance, in compression and tension tests of single-dowel corner joints with particle board (PB), Zhang and Eckelman [11] found that tension loads had a higher moment capacity than compression loads. Increasing the dowel diameter or length is directly correlated with enhanced bending strength. When evaluating the flexural strength of multi-dowel joints in PB, Zhang and Eckelman [12] concluded that a 75-mm spacing between dowels provided the highest moment capacity per dowel. Wan-Qian and Eckelman [13] conducted tests on screwed and dowelled corner joints under compression and bending, revealing the eccentric bending moment capacity. Their results demonstrated that the moment capacities of the fasteners significantly increased as long as the impact zones did not overlap. Smardzewski and Prekrat [14] examined the rigidity and strength of disconnected joints used in cabinet furniture, highlighting the importance of trapezoidal joints for optimal rigidity-strength properties, with wood dowels playing a significant role in supporting their strength. Tankut and Tankut [15] investigated the factors influencing the bending capacity of corner joints constructed with wood biscuits, emphasizing that joint strength mainly relied on edge gluing rather than the glued biscuits themselves.

In these tests, one crucial characteristic of the fasteners used was the minimal external energy required during assembly, making the process easy. The forces were typically applied using screws or eccentrics, such as screwing moments facilitated by screwdrivers [16,17]. Zhang et al. [18] examined the effects of screw diameter and length, loading type, board material, board surface condition, and gluing on the moment resistances of three-screw L-type corner joints under tension and compression loads. Their findings indicated that surfacing PB with synthetic resin and applying glue to the contact surfaces significantly improved the moment resistances of PB joints. The recommended screw diameter for connecting corner joints of case-type furniture was 5 mm with lengths of either 50 mm or 60 mm.

The rising popularity of flat-packed, ready-to-assemble (RTA) furniture has led to a need for creative approaches, particularly for individuals who may lack technical skills or have physical limitations. As a result, new innovative fasteners have been developed with original designs and user-friendly installation and disassembly methods. Computer simulations using the finite element method (FEM) have been employed to model these fasteners [19]. Furthermore, there are many studies in the literature in which the finite element method has been used in furniture engineering design.

In one study, the strength of a wooden chair was determined under different kinds of loads; then, the same chair was modeled and the stresses at some points under the same loads were determined using FEM. It was reported that the analysis data and experimental data were very similar [20]. In one study regarding using FEM in the structural analyses of furniture, it was difficult to analyze the stresses that occur in chair frames; but FEM could be used to solve this problem. In this research, a wooden chair frame was modeled and the analysis and design were demonstrated using FEM, and the analysis results were compared to the real test results. In conclusion, it was stated that chair frames can be analyzed using computer-aided structural analysis methods [21]. The difficulties in using FEM in wood materials have been compared and investigated with performance tests with finite element software [22]. Experimental tests and FEM analyses were performed for different types of sofa frames constructed of wood and wood-based materials. According to the results, the FEM results were given reasonable estimates regarding the strength properties of sofa frames. As a result, it was also emphasized that the joints are the critical points in furniture and that more durable joints can be made using materials with high bending strength [23].

Although conventional materials are typically used in furniture fastener production, there is an increasing interest and demand for smart materials. Negative Poisson’s ratio structures were described as early as 30 years ago [24,25,26], offering auxetic materials and structures with desirable mechanical properties, such as shear resistance, indentation resistance, synclastic behavior, varying permeability, high energy absorption, and fracture resistance [27,28]. Carneiro et al. [29] presented theories explaining the deformation behavior of auxetic materials, explored their mechanical properties, and showcased potential applications. Santulli and Langella [30] shared their experience using auxetic materials in various design objects, including chairs, bags, and seat belts. The structures were modeled as chiral with defined geometrical parameters, and real models were fabricated using neoprene or rubbery materials. Limited studies have explored the use of auxetics in the furniture industry. Smardzewski et al. [31] aimed to develop a model of an auxetic compression spring for seating furniture constructions. Ren et al. [32] designed, manufactured, and investigated the first auxetic nails for the wood and furniture industry, finding that auxetic nails did not consistently outperform non-auxetic ones. Tabacu and Stanescu [33] examined auxetic, anti-tetra-chiral structures designed as tubes subjected to tensile quasi-static loads, establishing theoretical calculation models to estimate the reaction force under tensile loads. Ren et al. [34] developed a simple tubular structure exhibiting auxetic behavior in both compression and tension by extending a recently proposed design concept for 3D metallic auxetic metamaterials. Zhang et al. [35] developed a combined tubular structure with tunable stiffness, improving bearing capacity and stability through the length design of the central column. Such design concepts have potential benefits for adaptive structures, smart devices, and applications in the furniture industry. A review of design methods and advanced manufacturing technologies for auxetic tubular structures can be found in [36]. The challenges and opportunities for applying auxetic tubes are discussed to inspire future research endeavors.

When reviewing the latest literature, it becomes apparent that several auxetic dowel designs have been developed for use in furniture joints, focusing on their mounting forces and withdrawal strengths [6,7]. These studies suggest that auxetic dowels could serve as an alternative fastener to traditional wooden dowels in furniture joints. However, the application of auxetic materials in furniture joints remains relatively limited. Therefore, this study aims to leverage the unique property of the negative Poisson’s ratio to design various types of auxetic dowels, facilitating easy insertion and enhanced resistance to pull-out forces in case furniture joints.

The aim of engineering design is to manufacture products in the ideal intersection of technical and economic considerations. Sometimes weak-strength products are strengthened, while sometimes, unnecessary excessive-strength products are reduced to a sufficient strength level, resulting in economic gain. The auxetic dowels within the scope of the study are designed to be used in corner joints of case-type furniture, and they can provide one-time ready-to-assemble (RTA) constructions. Furthermore, the auxetic dowels have advantages over the other fasteners commonly used in the furniture industry. These are:significantly lower cost,ease of assembly,reducing production operations and diversity,there is no need for any tools for assembly, it can be easily assembled by hand.

In this context, it could be said that the use of auxetic dowels as an alternative to traditional fasteners will provide significant technical and economic advantages to consumers and manufacturers. Accordingly, this study aims to design and produce different types of auxetic dowels using 3D printing technology and analyze the strength and stiffness of L-type corner joints connected with these dowels through theoretical, experimental, and numerical methods.

## 2. Materials and Methods

Figure 1 shows a flowchart illustrating the universality of the methodology used in this study.

In the methodology of the study; in the first stage, the design, analysis, and production of the auxetic dowels were carried out. In the second stage, the elastic properties of the produced dowels and the elastic and mechanical properties of the material (PA12) used in the production of the dowels were determined. In the next step, the strength and stiffness of the corner joints connected with the designed and manufactured dowels were experimentally, numerically, and theoretically analyzed. In the last stage, the results obtained from the experiments, numerical analyses, and theoretical calculations were compared and interpreted.

### 2.1. Design, Production, and Elastic Properties of the Auxetic Dowels and PA12

The purpose of the study was to design 8 types of auxetic dowels with different geometric patterns and 1 non-auxetic dowel with appropriate face (FM) and butt (BM) sizes as fasteners for corner joints in case furniture. The auxetic dowels were designed to include two types of inclusions (T: triangular and R: rectangular), two different inclusion sizes (A = 0.4 mm, B = 0.5 mm for rectangular inclusions, and A = 1 mm, B = 2 mm for triangular inclusions), and two different dowel hole diameters (3 and 4 mm). All of the designed dowels were 40 mm long and 8 mm in diameter; the corresponding butt muffs were 33 mm long and 12 mm in diameter, while the face muffs were 14.5 mm long and 12 mm in diameter. The general view of all types of auxetic surface dowels with different patterns and face and butt muffs is given in Figure 2, while the dimensions of the dowel and muffs are given in Figure 3.

For comparison purposes, a non-auxetic full dowel was designed as a reference dowel. The reference dowel was designed without inclusions on the surface (smooth surface), and there was no hole inside the dowel. Auxetic dowels were patterned with triangular and rectangular (rectangles with semicircles at two ends) inclusions of auxetic surfaces and with holes. The pattern on the entire dowel surface consisted of repeating each geometric inclusion configuration in the longitudinal and circumferential directions. The inclusion size factor has been described as the dimension of a single triangular or rectangular gap. All other dimensions depend on these gaps and their periodicity. Detailed inclusion sizes and cross-sections of the designed dowels are given in Figure 4.

3D models were first created using Autodesk Inventor v 2023.3 software during dowel production. Then, based on the CAD models, STP and STL models were prepared for numerical calculations and 3D printing. Finally, 3D printing selective laser sintering (SLS) technology was used in an EOS P396 printer (EOS GmbH, Munich, Germany) to produce the nine designed dowel types and corresponding muffs. The designed dowels were printed using polyamide EOS PA12—Polyamide 12 (EOS GmbH, Munich, Germany). It should be noted that the minimum link length of the designed auxetic dowel was only 1.1 mm, and this value approached the limit (1 mm) of the 3D printing company for manufacturing the dowels and muffs. The exact diameter of the dowel and the inner diameter of the muff was individually measured with a digital caliper. Uniaxial compression tests were performed on all dowel groups to calculate the coefficient of friction and the Poisson’s ratio of the dowels. In order to obtain the Poisson’s ratio of the dowels, a reference ruler was placed behind the dowels. Two pictures of the dowels were taken, one before loading and the other at the time of 2-mm deformation in the vertical (Y) direction. Then, dowel strains in vertical and horizontal directions were analyzed using National Instruments IMAQ Vision Builder v6.1 software (National Instruments, Austin, TX, USA). Poisson’s ratios were calculated by applying the edge detection method in the digital image analysis. Since the methodology for determining the coefficients of friction has been described in detail in [6,7], only the final results of the calculations in Table 1 are presented in this part of the paper. They were used for further numerical calculations.

In the study, we used eight types of auxetic dowels with triangular and rectangular inclusions and a reference non-auxetic dowel. The real pictures of the produced auxetic dowels are shown in Figure 5.

The mechanical and elastic properties of the PA12 material were determined in tensile tests according to the ASTM D3039/D3039M–17 [37] (Table 2). The dimensions and actual image of the tensile test specimen are shown in Figure 6.

Tensile samples were also printed using an EOS P396 printer (EOS GmbH, Munich, Germany) with a dimensional tolerance of 0.3 mm. A total of 10 samples were prepared. Tensile tests were performed on a numerically controlled Zwick 1445 universal testing machine (Zwick Roell AG, Ulm, Germany) with a 10 kN capacity. The loading rate was 10 mm/min. During the tests, strain and strain shortening at the center of the specimens were recorded using the Digital Image Correlation and Tracking method (DICT) using the Dantec system (Dantec Dynamics A/S, Skovlunde, Denmark). In order to include the plasticity properties of the material in numerical calculations, the experimental stress-strain relationship for polyamide (PA12) after exceeding the linear elastic range was determined (Figure 7). First, the linear elastic range was determined to establish the linear equation for this section. As shown in Figure 7, the slope of the line corresponds to the value of the modulus of linear elasticity for polyamide equal to E = 709 (standard deviation SD = 41) MPa, tensile strength MOR = 25 MPa (SD = 1.1 MPa), and Poisson’s ratio υ = 0.23 (SD = 0.01). Then, the true stress $\sigma_{T}$ and the logarithmic plastic strain $\varepsilon_{L}$, required in the finite element method (FEM) algorithm were calculated using the equations [6,7] given below:(1)$$\varepsilon_{L}=\varepsilon_{T}-\left(\frac{\sigma_{T}}{E}\right),$$
where $\sigma_{T}=\sigma_{e}\left(1+\varepsilon_{e}\right)$ true stress, $\varepsilon_{T}=ln\left(1+\varepsilon_{e}\right)$ logarithmic strain, E = modulus of elasticity of polyamide, $\sigma_{e}$ = engineering stress, and $\varepsilon_{e}$ = engineering strain. For the plastic range in Figure 7b above the straight line, the graph for $\sigma_{T}=f\left(\varepsilon_{L}\right)$ was plotted.

### 2.2. Mounting and Preparation of the L-Type Corner Joint Specimens

L-type corner joint specimens were prepared from three layers (Figure 8) of particleboard (PB) commonly used in case furniture construction. The PB materials were purchased from commercial suppliers. Some physical and mechanical properties of the PB used are presented in Table 2 [38].

The L-type specimens consisted of two structural members: a face member and a butt member. The face member measured 270 mm long, 150 mm wide, and 18 mm thick, while the butt member measured 270 mm long, 132 mm wide, and 18 mm thick. The members were joined together with two auxetic dowels without adhesive. The auxetic dowel corner connections provide a one-time ready-to-assemble (RTA) connection. The general configuration of the L-type corner joint specimens is shown in Figure 9.

During the assembly of the specimens, holes 12 mm in diameter and 14.5 mm deep were first drilled into the face piece and holes 33 mm deep were drilled into the butt piece of the specimens to insert the face and butt muffs. The face and butt muffs were then fully inserted and bonded to the holes using Jowat^®^ UniPUR 687.22 melamine-based adhesive (Jowat Swiss AG, Buchrain, Switzerland). Prior to assembly, the dowels were manually inserted into the muffs placed on the face members. A Zwick 1445 universal testing machine (Zwick Roell AG, Ulm, Germany) with a loading rate of 10 mm/min was used to determine the dowel assembly forces (Figure 10).

A total of 180 L-type corner joint specimens were prepared and tested, 10 per each type of joint. The experimental design of the study is shown in Table 3.

Prior to testing, corner joint specimens were stored for at least one month in a conditioning chamber at 20 °C ± 2 °C and 65% ± 3% relative humidity in order to avoid MC variations.

### 2.3. Tension and Compression Testing of the L-Type Corner Joint Specimens

Corner joints of case furniture are often subjected to compressive forces, which tend to open the joint, and tensile forces, which tend to close the joint, during usage. Bending moments occur at the corner joints under compression and tension loading conditions. In this study, these two important loading models were preferred as the test method to determine the strength and stiffness of L-type corner joints with auxetic dowels. Diagonal compression and tension tests of the L-type corner joint specimens were also performed on a numerically controlled Zwick 1445 universal testing machine (Zwick Roell AG, Ulm, Germany) with a 10 kN capacity at a loading rate of 10 mm/min under static loading. For the tensile tests, the bottoms of each of the two members of the joint were placed on the pieces with rollers on the bottom and V-shaped grooves on the top, allowing the two parts of the joint to outwardly move as the corner joint was loaded (Figure 11a). For the compression tests, two of the same V-groove pieces were placed at each end of the members. The specimen was then held by hand until the lower surface of the grip of the testing machine touched the upper surface of the upper V-groove pieces (Figure 11b). During the tension and compression tests, the maximum forces F (N) were measured to the nearest 0.01 N and deflection in the direction of the acting force D_F_ (mm) was determined to the nearest 0.01 mm. A total of 180 L-type specimens were tested; 90 were tested under tension, and the remaining 90 were tested under compression. First, the bending moment capacities ($M_{T}$, $M_{C}$) of the joints under tension or compression were calculated from the formulas:(2)$$M_{T}=0.5FL_{t}{}^{\prime }\;\left(\mathrm{Nm}\right)\;\mathrm{tension},$$
(3)$$M_{C}=FL_{c}{}^{\prime }\;\left(\mathrm{Nm}\right)\;\mathrm{compression},$$
where: F (N) is the maximum force of the joints, $L_{t}{}^{\prime }$ and $L_{c}{}^{\prime }$ (mm) are the length of the arm of the force F (N) under tension and compression, respectively. The moment arms ($L_{t}{}^{\prime },L_{c}{}^{\prime }$) were calculated as 0.09334 m and 0.08061 m for tension and compression, respectively (Figure 11). Then, the stiffness values of the corner joints were calculated as the quotient of the bending moment $M_{T}$ (Nm), $M_{C}$ (Nm), and the respective decrease or increase in the angle $\left(\varphi \right)$ (rad) a value between the joint arms.

These angles were determined based on the measurement of the deflection $D_{F}$ (mm) caused by the external load F (N). The stiffness coefficients $K_{T}$, $K_{C}$ (Nm/rad) for the joints under tension and compression are calculated by Equations (4) and (11), respectively:(4)$$K_{T}=\frac{FL_{t}{}^{\prime }}{\left(2\varphi \right)}$$
where:(5)$$\varphi =\left(\varphi_{2}-\varphi_{1}\right)$$
(6)$$L_{t}{}^{\prime }=\frac{\sqrt{2}}{2}(L_{B}-t)$$
(7)$$0.5\varphi_{1}=atg\left(\frac{L_{t}{}^{\prime }}{ƒ}\right)$$
(8)$$0.5\varphi_{2}=atg\left(\frac{L_{t}{}^{\prime \prime }}{ƒ-DF}\right)$$
(9)$$ƒ=L_{t}{}^{\prime }+\frac{\sqrt{2}}{2}t$$
(10)$$L_{t}{}^{\prime \prime }=\sqrt{{L_{t}}^{\prime 2}+{ƒ}^{2}-{\left(ƒ-D_{F}\right)}^{2}}$$
(11)$$K_{C}=\frac{FL_{c}{}^{\prime }}{\left(\varphi \right)}$$
where:(12)$$\varphi =\left(\varphi_{1}-\varphi_{2}\right)$$
(13)$$L_{c}{}^{\prime }=\frac{\sqrt{2}}{2}L_{B}-L_{c}{}^{\prime \prime }$$
(14)$$L_{c}{}^{\prime \prime }=t\sqrt{2}$$
(15)$$\varphi_{1}=2atg\left(\frac{\sqrt{2}L_{B}}{2L_{c}{}^{\prime }}\right)$$
(16)$$\varphi_{2}=2asin\left(\frac{\frac{\sqrt{2}}{2}L_{B}-D_{F}}{\sqrt{t^{2}+{\left(L_{B}-t\right)}^{2}}}\right)$$
were: $L_{B}$ (m) is the length of the butt member, and t (mm) is the thickness of the particleboard.

### 2.4. Theoretical Calculations for the Corner Joints

The tension and compression of hinges cause the mutual interaction of joints and arms at pivot points $P_{T},P_{C}$ (Figure 12a). The tension of the joints by bending moment $M_{T}$ (Nm) makes the indifferent axis pass through the point $P_{T}$, and the minimum and maximum normal stresses $\sigma_{min}$, $\sigma_{max}$ (MPa) occur nearer and further from the point $P_{T}$ (Figure 12b). Compression of the joints by bending moment $M_{c}$ (Nm) makes the indifferent axis pass through the point $P_{C}$, and the minimum and maximum normal stresses $\sigma_{min}$, $\sigma_{max}$ (MPa) occur closer and further from the point $P_{C}$ (Figure 12c). Therefore, it was decided to specify the values of these stresses for all types of dowels and joints used.

Figure 13 shows diagrams of the tension and compression of the corner joints. The normal stresses in the dowel of corner joints can be described in general form,
(17)$$\sigma =\frac{M_{i}}{\iint_{A}^{}y^{2}dA+{\left(0.5t\right)}^{2}A}y_{t}$$
where: $M_{i}$ (Nm) (i = T, C) bending moments as in Equations (2) and (3), $J_{o}=\iint y^{2}dA$ (m^4^) dowel moment of inertia, $y_{t}=\left\{\begin{array}{c} 0.5\left(t+d\right)\\ 0.5t\\ 0.5\left(t-d\right) \end{array}\right.$ (m) distance from the neutral axis, A (mm^2^) cross-section of dowel, d (mm) dowel diameter.

Hence, the maximum, minimum, and mean stresses in the dowels of joints under tension can be determined from the equations,
(18)$$\sigma_{max}=\frac{2\left(t+d\right)M_{T}}{{4J}_{o}+t^{2}A}$$
(19)$$\sigma_{min}=\frac{2\left(t-d\right)M_{T}}{{4J}_{o}+t^{2}A}$$
(20)$$\sigma_{mean}=\frac{2tM_{T}}{{4J}_{o}+t^{2}A}$$
assuming that,
(21)$$M_{T}={0.5F}_{y}L_{F}$$
(22)$$L_{F}=L_{B}-t$$
(23)$${0.5F}_{y}=0.5Fcos\left(\varepsilon -0.5\varphi \right)$$
(24)$${0.5F}_{x}=0.5F\sin\left(\varepsilon -0.5\varphi \right)$$
(25)$$0.5\varphi =\varepsilon^{\prime \prime }-\varepsilon^{\prime }$$
(26)$$\varepsilon^{\prime }=arccos\left(\frac{H}{L_{B}^{\prime }}\right)$$
(27)$$\varepsilon^{\prime \prime }=arcos\left(\frac{H-D_{F}}{L_{B}^{\prime }}\right)$$
(28)$$H=L_{T}+\sqrt{2}t$$
(29)$$L_{B}^{\prime }={\left(L_{B}^{2}+t^{2}\right)}^{0.5}$$
where: $F_{x},F_{y}$ (N) component Y and Y of external load F (N), $L_{F}$ (m) length of face member, $L_{B}$ (m) length of butt member, $D_{F}$ (m) deflection, ε (rad) inclination angle of face member, φ (rad) angle between arms of joint, and other parameters $\varepsilon^{\prime \prime },\varepsilon^{\prime }$ (rad), $H,L_{T},L_{B}^{\prime }$ (m) as shown in Figure 13a.

For the joints under compression, the maximum, minimum, and mean stresses in the dowels can be determined from the equations,
(30)$$\sigma_{max}=\frac{2\left(t+d\right)M_{c}}{{4J}_{o}+t^{2}A}$$
(31)$$\sigma_{min}=\frac{2\left(t-d\right)M_{c}}{{4J}_{o}+t^{2}A}$$
(32)$$\sigma_{mean}=\frac{2tM_{c}}{{4J}_{o}+t^{2}A}$$
where,
(33)$$M_{c}=F_{y}L_{F}$$
(34)$$F_{y}=F\cos\left(\varepsilon \right);\;F_{x}=F\sin\left(\varepsilon \right)$$
(35)$$\varepsilon =\beta -\varepsilon^{\prime }$$
(36)$$\beta =arcsin\left(\frac{L-0.5D_{F}}{L_{F}^{\prime }}\right)$$
(37)$$\varepsilon^{\prime }=arcsin\left(\frac{t}{L_{F}^{\prime }}\right)$$
(38)$$L_{F}^{\prime }={\left(L_{F}^{2}+t^{2}\right)}^{0.5}$$
where: ε (rad) inclination angle between arms of joints, other parameters $\beta ,\varepsilon^{\prime }$ (rad), $L,L_{F}^{\prime }$ (m) as shown in Figure 13b.

Considering the diverse shapes of the cross-sections of individual dowels, Table 4 shows the minimum cross-sectional area A (m^2^) and minimum moments of inertia $J_{o}$ (mm^4^) of the dowels according to the cross-sections.

Equations (9) and (21) were used to determine the relationships between the average stresses in the dowels and the deflections of the joints. The obtained results of the analytical calculations were compared with the results of the numerical calculations.

### 2.5. Numerical Model of the L-Type Corner Joints

In order to investigate the damage mechanism of the auxetic dowels under the tension and compression loading of corner joints, the finite element method was used to simulate the deflection, deformation, and failure process of the fasteners. According to the symmetry, half of the joints were constructed with the dimensions of the tested specimens, as shown in Figure 8 and Figure 9, using the commercial software ABAQUS/Explicit v6.14 (Dassault Systemes Simulia Corp., Waltham, Ma, USA). A nonlinear model was selected for the calculations to account for the geometric and material nonlinearity of each component. The elastic properties of PB and PA12 at the limits of linear elasticity are given in Table 2. In addition, the plastic properties of PA12 are shown in Figure 6. For polyamide, the ductile damage model was adopted with a fracture strain of 0.03, a stress triaxiality of 1/3, a strain rate of 1, and a displacement at failure of 0.1. As shown in Figure 14, the face and butt members of the corner joints were modeled by C3D8R—eight nodes, linear brick, reduced integration, and hourglass control elements (total number of nodes: 16642; total number of elements: 13324). The maximum mesh size of the arms was about 4 mm.

The optimum face and butt member mesh size was selected based on a series of numerical calculations. Figure 15 shows the effect of the mesh size on the load value. This figure shows that satisfactory agreement between the numerical results of the experimental tests can be obtained for a 4 mm mesh and a computation time of about 8 h. A mesh size of 2 mm obtained similar results. but over a much longer time. Contact with a friction coefficient of 0.1 was added between the arms of the joint.

Models of dowels and muffs were created according to the nominal dimensions shown in Figure 1. The element type of the dowels was C3D10: a 10-node quadratic tetrahedron (number of nodes: 61732, number of elements: 38239), with an approximate global element size of less than 0.5 mm. A series of numerical calculations tested the mesh convergence analysis for the dowel and muff until the results satisfactorily converged. The convergence analysis showed that the most preferred numerical models were obtained when the inclusions were omitted along the entire length of the dowel and only used at the point of contact between the two arms (Figure 16). In Abaqus, dowels were partitioned into three parts: a solid shorter part, an inclusions part and a solid longer part. The mechanical properties of PA12, as in Table 2, were used for the inclusions part. In the case of the solid parts, mechanical properties of PA12, as in Table 2, were used, but negative Poisson’s ratios were applied from Table 1 for each dowel type. A surface contact with a coefficient of friction, as shown in Table 1, was applied between the dowel and muff.

As a result, a model corresponding to the real model was obtained, which showed differences in the diameters of the muffs and dowels. It was also decided to use a rigid “tie”-type connection between the muffs, butt, and face members [6]. This jointing method is equivalent to gluing a muff into a particleboard socket.

The crosshead and support were set to be rigid bodies and modeled by R3D4—four-node 3D bilinear rigid quadrilateral element. The contact interaction property was defined in two aspects between sphere, crosshead, and support. First, the “hard” contact was defined for all of the normal behavior, and a friction coefficient of 0.1 was used for tangential behavior. At the reference point (RP-T), a vertical deflection equal to 10 mm was applied. Measurements of the force and deflection were made every 0.05 s. Computations were performed at the Poznań Supercomputing and Networking Center (PSNC).

## 3. Results and Discussion

### 3.1. Mounting Force Results of L-Type Corner Joints

One of the hypotheses of this study was that the auxetic dowels would require less mounting force during the assembly operations and, therefore, would be easier to assemble compared with the non-auxetic dowels. This hypothesis was based on the fact that the diameter of auxetic dowels decreases under the mounting forces, which is an extraordinary property related to the negative Poisson’s ratio. Accordingly, the mounting forces of the L-type corner joint specimens were obtained from the tests and are presented in Figure 17. At the same time, it was recognized that, due to the apparent differences between the results, it was not necessary to perform a statistical analysis of the significance of the differences. Therefore, only the mean values and coefficients of variation were presented in the figure.

Looking at the assembly forces used to assemble the L-type corner joint specimens, the corner joints assembled with auxetic dowels had lower assembly forces than the corner joints assembled with reference dowels for all groups. The corner joint specimens connected with reference dowels required a mounting force of 990 N, while the corner joint specimens connected with auxetic dowels (R3A, R3B, R4A, R4B, T3A, T3B, T4A, and T4B) could be assembled with mounting forces of 469 N, 357 N, 650 N, 624 N, 379 N, 300 N, 618 N, and 503 N, respectively. Accordingly, it can be said that the diameters of auxetic dowels were reduced under the mounting forces, and therefore, they could be assembled with lower mounting forces. In other words, the results of the corner joint mounting force tests showed that the hypothesis of the study was accepted, and the required mounting force of the corner joints connected with the auxetic dowels was significantly lower than the non-auxetic dowels.

When the mounting forces were compared according to the dowel holes and inclusion size, increasing the hole diameter significantly increased the required mounting forces, while increasing the inclusion size slightly decreased the required mounting forces.

### 3.2. Experimental Results for the Strength and Stiffness of L-Type Corner Joints

In the tension and compression tests, a failure mode was observed for all groups in the form of dowel breakage without particleboards for all groups. Both tension and compression tests lasted approximately 60–90 s. It can be said that the type of failure is completely dependent on the inclusion size and the diameter of the dowel hole. This phenomenon can be explained by the fact that the inclusions and holes reduce the cross-sectional area of the dowels. During the tests, the normal and shear stresses occurring in the cross-sections of the dowels under tensile or compressive forces exceeded the stress limits of the PA12 material (MOR = 25 MPa). As a result, the dowels broke before they were pulled out of the muff. The actual pictures of the type of failure of corner joints with auxetic dowels and the cross-sectional areas of all auxetic dowels are shown in Figure 18.

In the tests, the corner joints connected to the dowels with triangular inclusions gave very low load capacity results under both loading conditions. Particularly, under the compressive loads, these corner joints broke before they could bear any load; i.e., under a mass of joint arms. The characteristic relationships obtained during the tests between the applied load and deflection and the relationship between the stiffness coefficient and the load of the tested corner joints are shown in Figure 19.

Characteristic ranges of joint stiffness could be given as follows; (I)—linear stiffness range, (II)—plastic stiffness range, (III)—digressive stiffness range, (IV)—semi-constant stiffness range, and points of the joint stiffness: $K_{e}$—stiffness for the proportional range, $K_{m}$—stiffness for the plastic range, maximum stiffness, $K_{c}$—the beginning of the semi-constant stiffness. The results of the load-deflection relationship and the values of the load-bearing capacity obtained from the corner joints in the tension and compression tests are shown in Figure 20. 

According to the load-deflection relationships, in general, the corner joints connected with RF dowels are characterized by a large angle of curves to the horizontal axis until the failure loads for both tests. For the joints connected with auxetic dowels, the obtained smaller angle of curves with respect to the horizontal axis indicates the lower strength and stiffness of these corner joints compared to the joints connected with RF dowels. Figure 20a,b shows that the corner joints connected with auxetic dowels with rectangular inclusions showed similar behavior to each other and to the corner joints connected to RF dowels, but the corner joints connected to dowels with triangular inclusions showed different behavior under both loads. It can also be observed in Figure 20a,b that curves for the corner joints under both compression and tension loadings are smooth with no rapid changes in the load values. In the case of the corner joints connected with rectangular dowels, the maximum load capacity values are achieved up to approximately 3–4 mm of deflection under tension, except for R4A (6 mm). After this point, the load capacity values of the corner joints become lower. Under compressive loading, for the corner joints connected to the dowels with rectangular inclusions, the deflections at which the maximum load was reached differed according to the dowel groups. It was about 2 mm for the R4A dowels, 6 mm for the R3B dowels, and 4 mm for both the R3A and R4B dowels.

The basic parameter for the strength of corner joints is provided by the maximum load at the failure of specimens under tension or compression tests. According to Figure 20c,d, in general, corner joints connected with the auxetic dowels gave much lower load capacity values than the corner joints connected with non-auxetic dowels under both tension and compression loads. Among the corner joints connected with auxetic dowels; the joints connected with the R3B and R4B dowels gave the highest results for both under tension (28.08 N, 27.07 N) and compression (9.36 N, 9.27 N), respectively. It can be seen from the results that the dowels with rectangular inclusions gave higher load capacities than the dowels with triangular inclusions. The corner joints connected with auxetic dowels with triangular inclusions (T3A, T3B, T4A, T4B) gave very low values of load-bearing capacity (5.15 N, 4.34 N, 9.55 N, 3.41 N) in terms of tension, while they could not carry any load in compression, but T3B dowels could. Acceptable results could not be obtained from the dowels with triangular inclusions. In the case of the corner joints connected with rectangular inclusions (R3A, R3B, R4A, R4B), it can be said that the average load-bearing capacity values of corner joints increased as the inclusion size increased or as the dowel hole diameter decreased both in tension (24.09 N, 28.01 N, 17.70 N, 27.07 N) and in compression (7.48 N, 9.36 N, 5.14 N, 9.27 N). Under the tensile or compressive loads, the dowels were subjected to considerable normal and shear stresses, and the cross-sectional areas of these auxetic dowels could not support the normal and shear stresses. In addition, increasing the diameter of the dowel hole significantly reduced the cross-sectional area, and these dowels became even weaker. The fact that the failure mode observed in the tests of these couplings was a dowel fracture confirmed this situation. The grouping data for the corner joints tested under tension yielded a mean load capacity of 14.92 N, while the grouping data for the corner joints tested under compression resulted in a mean load capacity of 6.76 N. Therefore, in general, it can be concluded that the corner joints loaded under tension have approximately two times greater load capacities than those loaded under compression.

The results of the stiffness-load relationship and the $K_{e}$, $K_{m}$, and $K_{c}$ values for the joints during tension and compression tests are given in Figure 21. As seen from Figure 21a,b, depending on the type of loading, the stiffness-load relationships differed in terms of mechanical behavior properties. However, when individually analyzed for both types of loading, the corner joints connected with auxetic dowels and RF dowels showed similar mechanical behavior characteristics in terms of stiffness-load relationships. For tension loads, the stiffness-load relationships were initially linear to a point, and then became curvilinear relationships, and finally continued almost flat and parallel to the horizontal axis. In the case of compression loads, the stiffness-load relationships were also initially linear up to the maximum stiffness values, after which a sharp decrease was observed, and finally continued almost parallel to the horizontal axis.

The stiffness of the corner joints was evaluated based on the changes in the stiffness coefficient K (Nm/rad) as a function of the angle of rotation φ (rad) between the joint arms. According to Figure 21c,d, it can be clearly seen that the corner joints connected with the RF dowels gave much higher stiffness values than the corner joints connected with auxetic dowels under both tension and compression loads. The corner joints connected with auxetic dowels were found to be quite flexible compared to the corner joints connected with RF dowels. Among the corner joints connected with auxetic dowels, the corner joints connected with the R3B and R4B dowels gave the highest results under tension (599 Nm/rad, 589 Nm/rad) as well as strength, while the corner joints connected with R3A dowels gave the highest stiffness values under compression (470 Nm/rad). The stiffness values of the dowels with rectangular inclusions gave higher stiffness values than the dowels with triangular inclusions. In the case of the corner joints connected with the dowels with rectangular inclusions (R3A, R3B, R4A, R4B), it can be said that the mean stiffness values of corner joints increase as the inclusion size increases, or the dowel hole diameter decreases (460 Nm/rad, 599 Nm/rad, 376 Nm/rad, 589 Nm/rad) under tension loads. For compressive loads, the stiffness values obtained were 470 Nm/rad, 266 Nm/rad, 313 Nm/rad, and 286 Nm/rad, respectively, for the corner joints connected with the dowels with rectangular inclusions. The corner joints connected with auxetic dowels with triangular inclusions (T3B, T4A, T4B) gave very low stiffness values (231 Nm/rad, 306 Nm/rad, 319 Nm/rad) in tension, except for the T3A dowel (451 Nm/rad); while they could not carry any load in compression, except for the T3B dowel (298 Nm/rad). Overall, it could be said that acceptable stiffness values could not be observed for the dowels with triangular inclusions. For the auxetic dowels, the stiffness values were very close between the corner joints connected with the R3A (460 Nm/rad) and T3A (451 Nm/rad) dowels and the corner joints connected with the T4A (306 Nm/rad) and T4B (319 Nm/rad) dowels under tensile loading. Similarly, very close stiffness values were obtained from the corner joints connected with R4B (286 Nm/rad) and T4B (298 Nm/rad) dowels under compressive loads.

### 3.3. Comparison of the Experimental Results, Numerical Analyses, and Theoretical Calculations

To provide a practical evaluation of how well the maximum loads and deflections obtained from numerical analyses (FEM) agree with the observed maximum loads and deflection results from the actual tests, comparisons of the observed test results with the FEM results are shown in Table 5. This table comparatively summarizes the maximum failure loads and deflections experimentally and numerically obtained, and the stiffness values with their coefficients of variations.

Table 5 shows that, with the exception of a few dowel groups, the experimental and numerical values are in good agreement for both tension and compression tests. In general, it can be seen that the load and deflection values of the corner joints connected to the dowels with rectangular inclusions can be better predicted than the corner joints connected to the dowels with triangular inclusions.

The numerical and experimental load-deflection relationships of corner joints connected with RF and auxetic dowels under tension and compression loads are shown in Figure 22 for each group. The comparisons shown are not for average values, but for individual selected connections from each group. In this way, it was ensured that the numerical model corresponds to a specific dowel and a specific muff. For this reason, the strength values discussed will differ from the mean values.

In the case of the corner joints connected with RF dowels (Figure 22a,b), the numerical (FEM) results presented very close mechanical behavior and strength values to the actual test results of the corner joints according to the load-deflection relationships. In general, for the auxetic dowel-connected corner joints, it can be said that the FEM results also gave reasonable estimates for the mechanical behavior properties of the corner joints.

It can be seen from the results that for the corner joints connected to the dowels with rectangular inclusions (Figure 22c), the differences between the FEM and actual test results were very small for R3B and R4B dowels, while they were relatively larger for R3A and R4A dowels under tension. Accordingly, it was observed that the reliability of the numerical analyses decreased with the increase in the inclusion size for the dowels with rectangular inclusions under tension. However, in the case of the compression tests (Figure 22d), as can be seen, the FEM and actual test results are quite consistent for all types of auxetic dowels with rectangular inclusions.

Regarding the corner joints connected with auxetic dowels with triangular inclusions (T3A, T3B, T4B), very close mechanical behavior and strength values were obtained in terms of the FEM and actual test results, except for the T4A dowels (Figure 22e) under tension loads. In the actual compression tests (Figure 22f), the corner joints connected with the auxetic dowels with triangular inclusions failed before they could carry the load, but the T3B dowels did. Therefore, only FEM results were available for the corner joints connected to these (T3A, T4A, T4B) dowels. However, for the corner joints connected with T3B dowels, the FEM and actual test results were consistent in terms of mechanical behavior and strength values.

The failure models and normal stress distribution in the auxetic dowels with rectangular and triangular inclusions, respectively, under tension and compression loads are shown in Figure 23.

According to the numerical analyses (Figure 23), it can be clearly seen that the maximum stresses are concentrated near the inclusions and increase for both rectangular or triangular inclusions of the auxetic dowels during tensile and compressive loadings. It should be noted that the above figure shows the fact that ABAQUS has removed all damaged elements in the calculation. The highest stress values were obtained in dowels with rectangular inclusions. As shown in Table 4 and Figure 18, this type of dowel is characterized by a higher cross-sectional area and moment of inertia than dowels with triangular inclusions. The maximum stresses were observed in dowels R3A (27.67 MPa), R3B (27.67 MPa), R4A (24.35 MPa), and R4B (25.91 MPa), which carried the highest loads in the tension tests (Figure 23a). In the compression test, the stresses were lower for R3A (5.13 MPa), R3B (5.63 MPa), R4A (2.67 MPa), and R4B (4.27 MPa). This tendency is emphasized in numerous scientific works and mainly reflects the lower bending moments in the case of compressed joints [15,39,40,41]. For dowels with triangular inclusions, the mean normal stress also depends on the cross-sectional area and bending moments. Under tension, (Figure 23b) normal stress was equal to T3A (7.35 MPa), T3BA (7.48 MPa), T4A (9.04 MPa), and T4B (7.81 MPa), respectively. In the compression test, the stresses were generally lower, about 1–2 MPa. Based on these results, it can be observed that the dowels with rectangular inclusions yielded higher stresses than the dowels with triangular inclusions, and all groups of auxetic dowels have higher stress values under tension than under compression loads.

Trends in the results of numerical calculations also confirm the results of the analytical analysis. Figure 24 shows the relationships between the theoretically calculated mean normal stress values and the deflections of the corner joints. For dowels with rectangular inclusions (Figure 24a), the maximum stresses in tension are 25.9 MPa (R3A), 24.6 MPa (R3B), 19.9 MPa (R4A), and 25.2 MPa (R4B), and in compression (Figure 24b) they are 4.58 MPa (R3A), 4.88 MPa (R3B), 2.9 MPa (R4A), and 4.84 MPa (R4B), respectively. Comparing these values with the results of the corresponding numerical calculations for R3A, R3B, R4A, and R4B, it can be seen that the numerical values are higher in tension by 6.4%, 11.1%, 18.3%, and 2.7%, and in compression by 10.7%, 13.3%, −8.6%, and −13.3% (minus means a decreasing tendency). For dowels with triangular inclusions, only the tensile test results were compared. The analytical value of normal stress was 7.8 MPa for T3A-type dowels and 8.09 MPa (T3B), 13.05 MPa (T4A), and 4.1 MPa (T4B) for other dowels. A comparison of these values with the results of corresponding numerical calculations shows that numerical values are lower by 6.1%, 8.2%, 44.4%, and −47.5%, respectively (minus means growing tendency).

As a result of the comparisons, it was observed that there is consistency between the mean stress values obtained as a result of theoretical calculations and the numerical analysis. In addition, the good compatibility of the obtained results of the numerical calculations with the experimental results confirms a sufficient correct calibration of the numeric model. In addition, the compatibility of numerical calculations with analytical results also indicates that the analytical model is sufficiently accurate.

## 4. Conclusions

The analysis of the test results showed that the mounting force for assembling the corner joints connected with the auxetic dowels was significantly lower than that for the corner joints connected with non-auxetic dowels. Thus, furniture users can assemble furniture more easily and without using tools. However, based on the experimental and numerical calculations, the strength and stiffness of the corner joints connected with the reference dowels were significantly higher than those connected with the auxetic dowels. Furthermore, theoretical and numerical analyses showed that the dowels used to connect the corner joints were subjected to a considerable amount of normal and shear stresses under both tension and compression loads. The failure modes of dowel fracture obtained from the tests also confirmed this phenomenon. Therefore, from a research perspective, new structures of dowels or tubes should be developed that will have higher cross-sections, auxetic properties, and greater bending strength. However, an important observation is that corner joints connected with the dowels with rectangular inclusions gave much better results than the dowels with triangular inclusions. Acceptable results could not be obtained from the dowels with triangular inclusions. For the corner joints connected to dowels with rectangular inclusions, the strength and stiffness of the corner joints increased as the size of the inclusions increased, or as the diameter of the dowel hole decreased. These suggestions may inspire further research on the modification of rectangular inclusions. In conclusion, it could be said that the auxetic dowels with rectangular inclusions should be improved before being used as an alternative fastener for traditional furniture dowels in the engineering design approach. Therefore, it is suggested that, in future studies, the auxetic dowels should be produced with more robust materials and/or different 3D printing or injection technologies should be tried for the production of the dowels.

## Figures and Tables

**Figure 1 materials-16-04547-f001:**
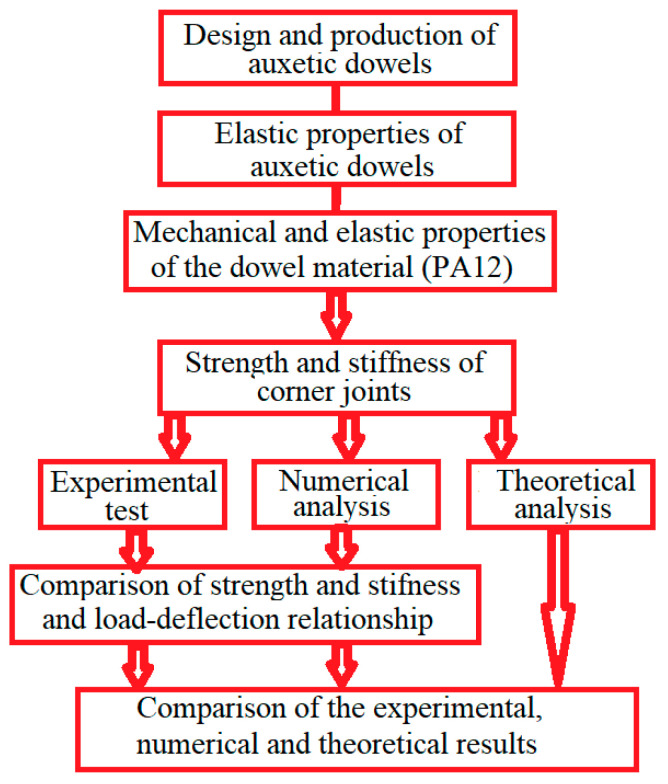
The flowchart of the used methodology.

**Figure 2 materials-16-04547-f002:**
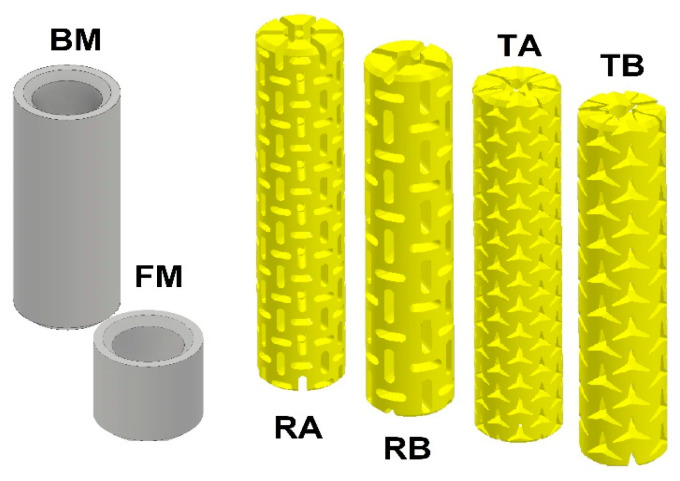
General view of the muffs and designed auxetic surface dowels.

**Figure 3 materials-16-04547-f003:**
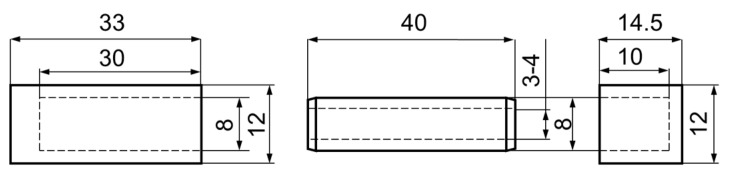
Dimensions of the dowel and appropriate muffs.

**Figure 4 materials-16-04547-f004:**
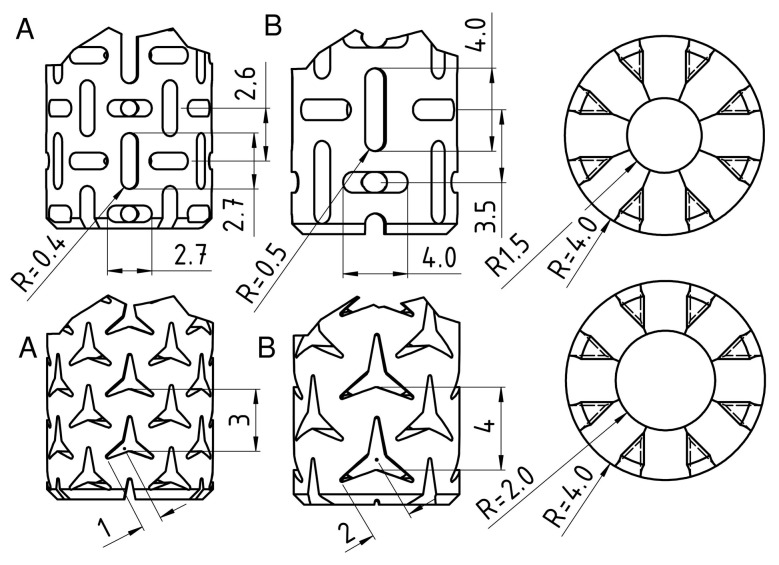
Geometric details of typical rectangular inclusions, triangular inclusions, and the cross-sectional geometry of auxetic dowels.

**Figure 5 materials-16-04547-f005:**
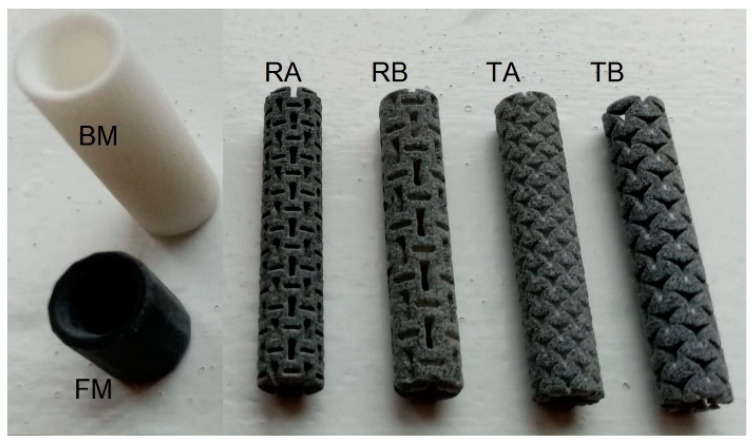
Face (FM) and butt (BM) muffs and all types of produced dowels with PA12.

**Figure 6 materials-16-04547-f006:**
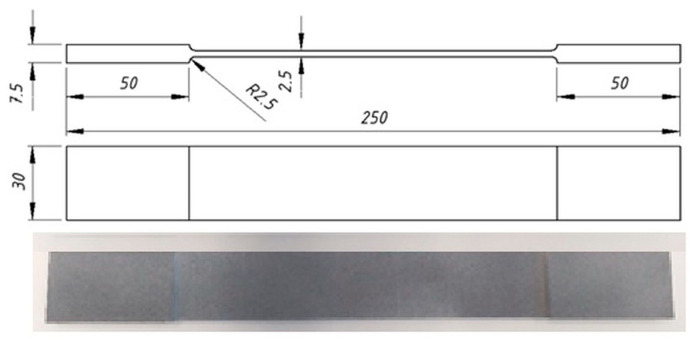
Front view, side view, and the real picture of the tensile test specimens (in mm).

**Figure 7 materials-16-04547-f007:**
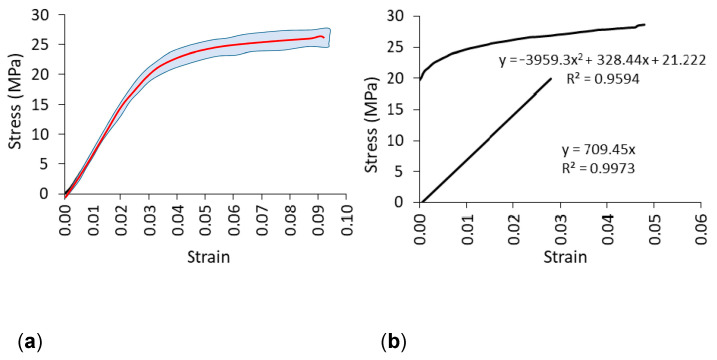
Stress-strain relationship for PA12; original curve from the experiment (**a**), limits of linear elasticity and plasticity (**b**).

**Figure 8 materials-16-04547-f008:**
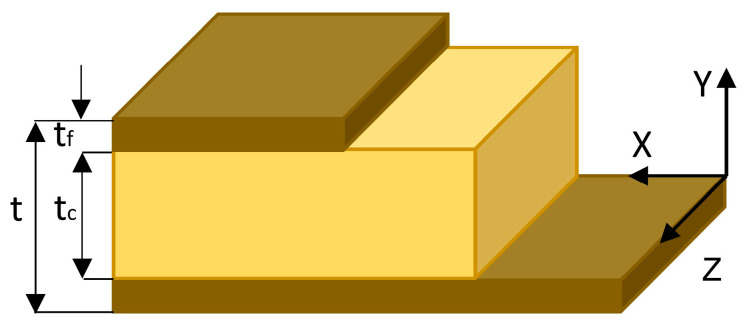
Three layers of PB used for preparing the specimens in this study.

**Figure 9 materials-16-04547-f009:**
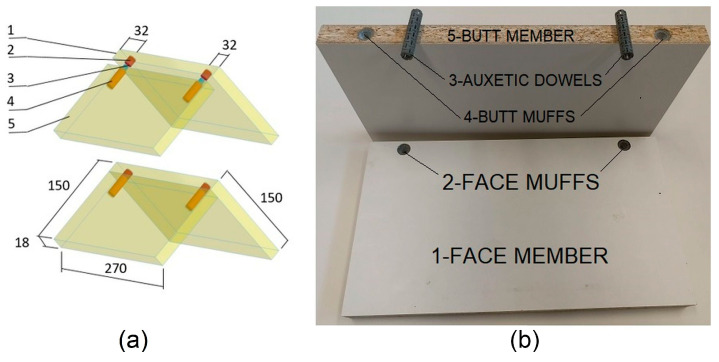
Dimensions (**a**) and general configuration (**b**) of the corner joint specimens.

**Figure 10 materials-16-04547-f010:**
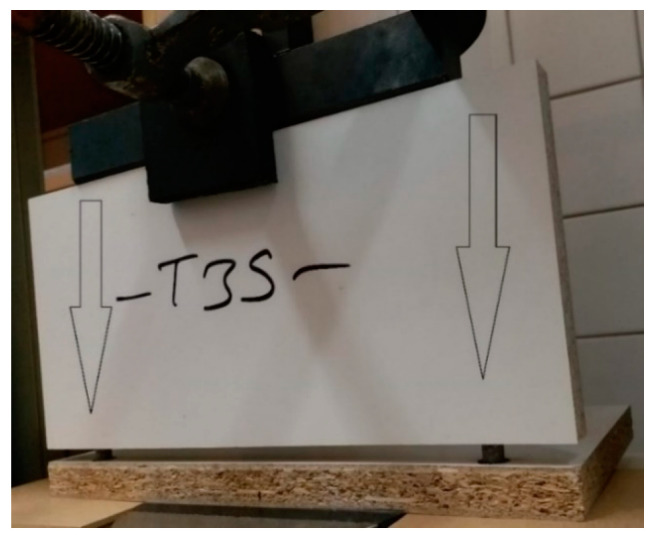
Mounting of the L-type corner joint specimens.

**Figure 11 materials-16-04547-f011:**
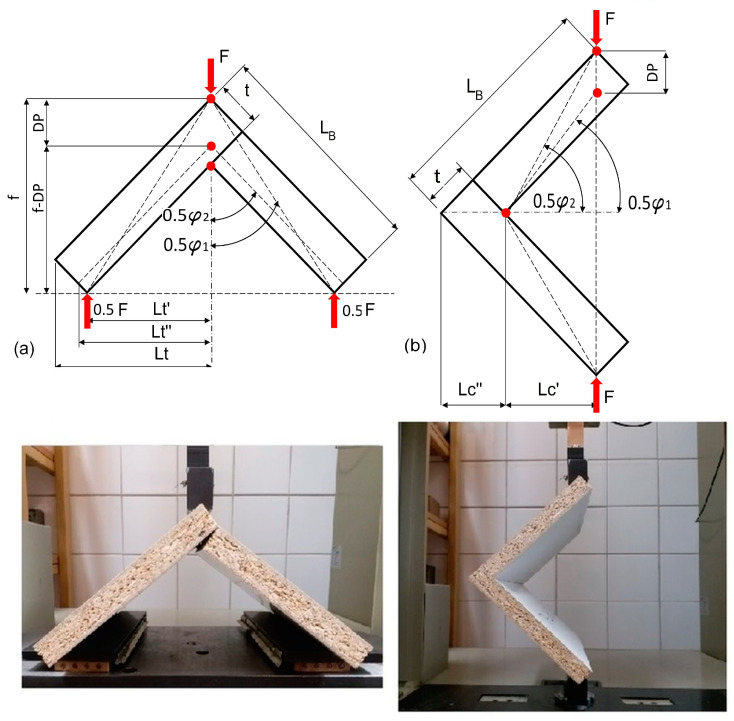
Loading of the corner joints: (**a**) under tension, (**b**) under compression.

**Figure 12 materials-16-04547-f012:**
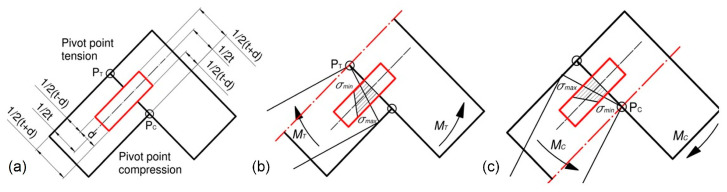
Characteristic pivot points (**a**), the normal stresses in dowels under tension (**b**), and compression (**c**) for the L-shaped joint specimens.

**Figure 13 materials-16-04547-f013:**
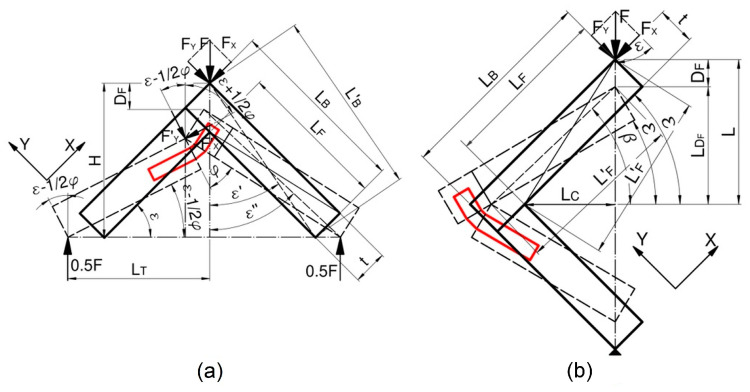
Static diagram of the tension (**a**) and compression (**b**) of joints.

**Figure 14 materials-16-04547-f014:**
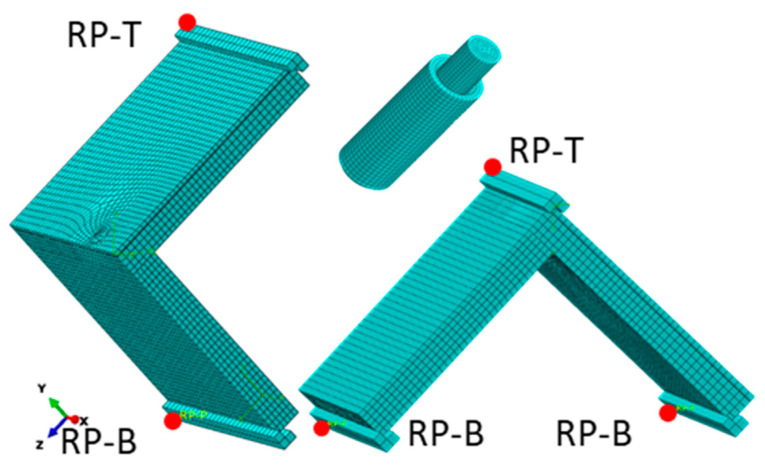
Mesh models of joints where red points: RP-T, RP-B—a reference points for the top and bottom support, respectively.

**Figure 15 materials-16-04547-f015:**
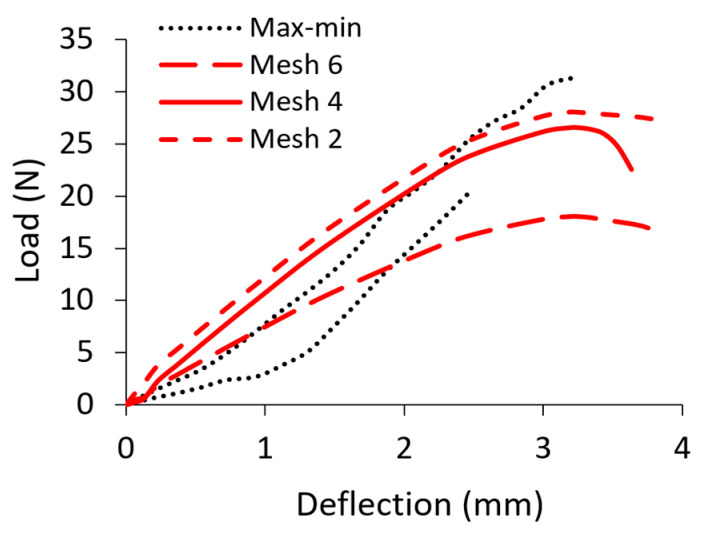
Mesh size effect on the results of the numerical calculation for R4B.

**Figure 16 materials-16-04547-f016:**
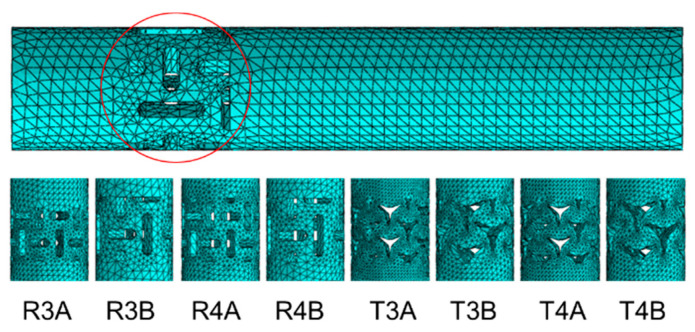
Mesh models of dowels.

**Figure 17 materials-16-04547-f017:**
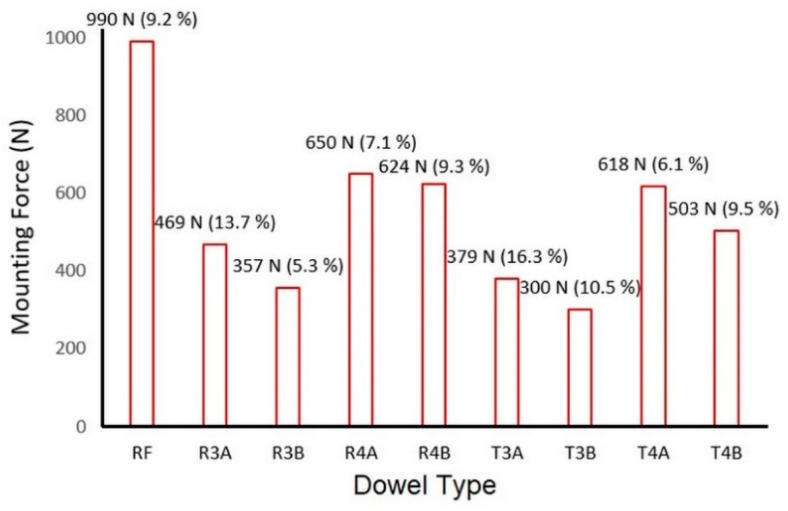
Mounting force results of the non-auxetic and auxetic dowels (values in parentheses are the coefficients of variation).

**Figure 18 materials-16-04547-f018:**
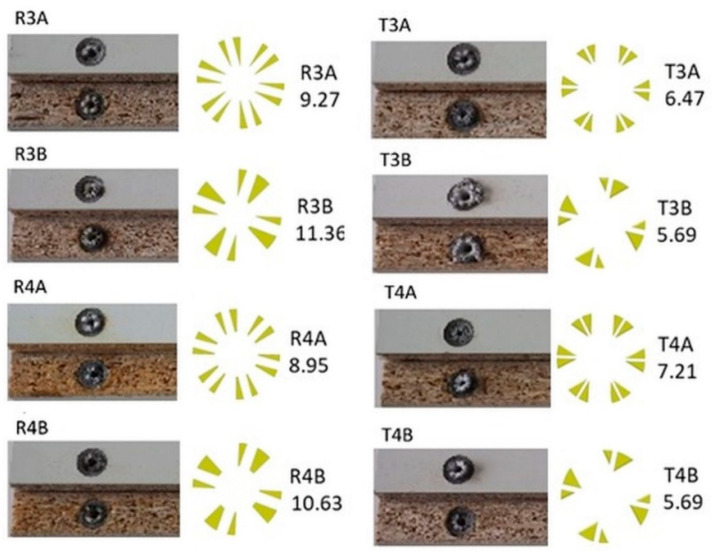
Failures of auxetic dowels under tension or compression, and cross-sectional area of the dowels depending on the inclusion size and hole diameter (in mm^2^).

**Figure 19 materials-16-04547-f019:**
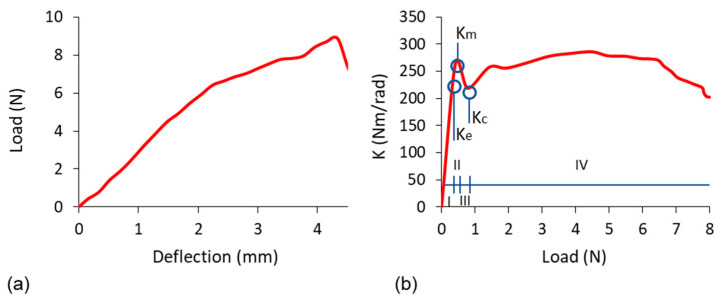
Characteristic relationship between the load and deflection (**a**) and the relationship between the stiffness coefficient and load (**b**) of the tested joints. Characteristic ranges of joint stiffness were shown by circles: (I)—linear stiffness range, (II)—plastic stiffness range, (III)—digressive stiffness range, (IV)—semi-constant stiffness range, and points of the joint stiffness.

**Figure 20 materials-16-04547-f020:**
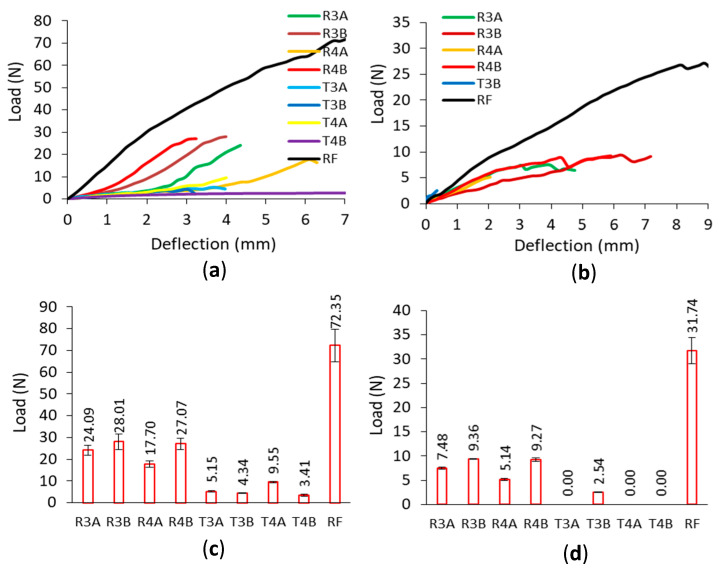
Relationship between load-deflection (**a**,**b**) and load capacities (**c**,**d**) for the joints under tension and compression, respectively (whiskers illustrated standard deviations).

**Figure 21 materials-16-04547-f021:**
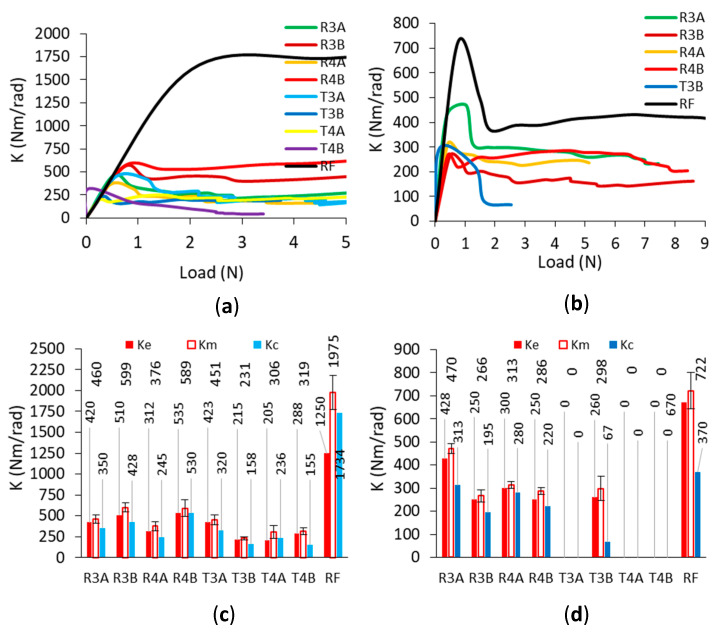
Relationship between stiffness and load (**a**,**b**) and $K_{e}$, $K_{m}$, and $K_{c}$ (**c**,**d**) for the joints under tension and compression (whiskers illustrated standard deviations).

**Figure 22 materials-16-04547-f022:**
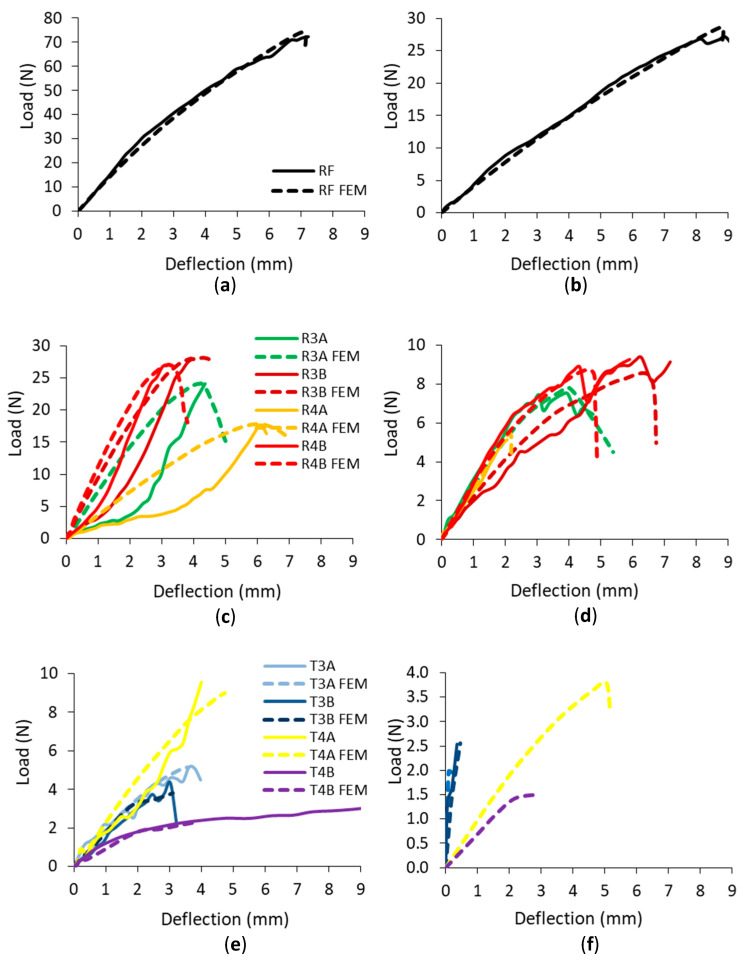
Relationship between load and deflection for the corner joints connected with RF and auxetic dowels under tension (**a**,**c**,**e**) and compression (**b**,**d**,**f**).

**Figure 23 materials-16-04547-f023:**
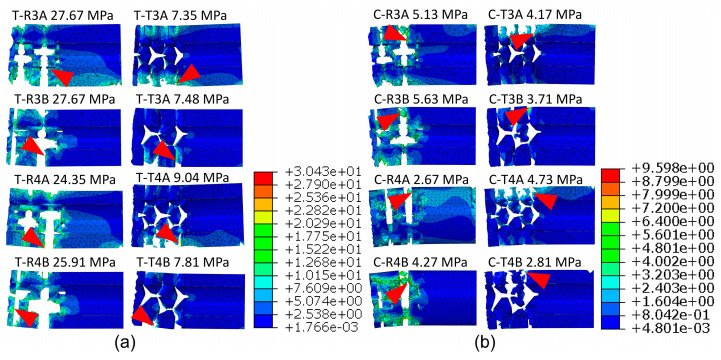
Normal stress distribution in auxetic dowels with rectangular (R) and triangular (T) inclusions under tension (T, **a**) and compression (C, **b**). Red triangles indicate the location of maximum stresses.

**Figure 24 materials-16-04547-f024:**
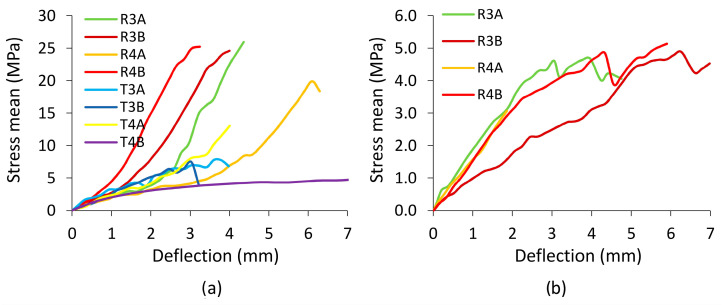
The relation between theoretically calculated mean normal stress in the dowel and the joint deflections under tension (**a**) and compression (**b**), respectively.

**Table 1 materials-16-04547-t001:** Characteristics of designed and produced auxetic and non-auxetic dowels in this study.

Dowel (Inclusion) Type	Dowel Hole Diameter (mm)	Inclusion Size (mm)	Mean Diameter of Dowels (mm)	Mean Inner Diameter of Muffs (mm)	Mean Tolerance (mm)	Poisson’s Ratio of Dowel (ϑ)	Friction Coefficient of Dowel	Dowel Code
Non-auxetic	RF	Plain	7.86	7.59	+0.27	0.201 *	0.033	RF
Triangular	3	A (1)	7.91	7.56	+0.35	−0.208	0.333	T3A
		B (2)	7.90	7.56	+0.34	−0.300	0.212	T3B
	4	A (1)	7.89	7.54	+0.35	−0.352	0.416	T4A
		B (2)	7.89	7.49	+0.39	−0.391	0.345	T4B
Rectangular	3	A (0.4)	7.87	7.58	+0.29	−0.356	0.367	R3A
		B (0.5)	7.86	7.52	+0.35	−0.367	0.244	R3B
	4	A (0.4)	7.86	7.59	+0.27	−0384	0.458	R4A
		B (0.5)	7.87	7.55	+0.32	−0.445	0.390	R4B

* Non-auxetic.

**Table 2 materials-16-04547-t002:** Physical and mechanical properties of the layers of the PB [38].

Property	Unit	PB			PA12
		t = 18	t*_f_* = 3	t*_c_* = 12	
MC	%	6.18 (0.08 *)	5.2 (0.09)	6.54 (0.1)	(-)
D	kg/m^3^	649 (7)	882 (4.8)	541 (5)	938 (140)
υ		0.29			0.23 (0.042)
G	MPa	991 (112)	1298 (85)	682 (72)	288
$E_{x}$		2556 (290)	3350 (220)	1760 (185)	709 (41)
MOR		10.9 (1.8)	14.1 (2.1)	7.8 (1.7)	25.8 (1.1)

* Values in parenthesis standard deviations, MC: Moisture content, D: Density, υ: Poisson’s ratio, G: Modulus of rigidity, $E_{x}$: Modulus of elasticity, MOR: Modulus of rupture.

**Table 3 materials-16-04547-t003:** Experimental design of the study.

Dowel (Inclusion) Type	Dowel Hole Diameter (mm)	Inclusion Size (mm)	Tension Test		Compression Test	
			Specimen Replication	Number of Dowel	Specimen Replication	Number of Dowels
Non-auxetic	Non-hole	Plain	10	20	10	20
Triangular (T)	3	A (1)	10	20	10	20
	3	B (2)	10	20	10	20
	4	A (1)	10	20	10	20
	4	B (2)	10	20	10	20
Rectangular (R)	3	A (0.4)	10	20	10	20
	3	B (0.5)	10	20	10	20
	4	A (0.4)	10	20	10	20
	4	B (0.5)	10	20	10	20
Total (Auxetic + non-auxetic)			80 + 10 = 90	160 + 20 = 180	80 + 10 = 90	160 + 20 = 180
Total (Specimens/dowels)			180 Specimens/360 Dowels			

**Table 4 materials-16-04547-t004:** Cross-sectional area and moment of inertia values of the dowels.

Dowel Type	A (mm^2^)	$J_{o}$ (mm^4^)
R3A	9.27	49.54
R3B	11.36	59.31
R4A	8.85	48.61
R4B	10.63	57.57
T3A	6.47	40.90
T3B	5.69	36.47
T4A	7.21	44.89
T4B	5.69	36.47

**Table 5 materials-16-04547-t005:** Differences between the maximum failure loads and deflections for the results of experiments and numerical analyses, and stiffness values for tension and compression tests.

Dowel Type	Experimental Results			FEM Results		Difference for Loads (%)	Difference for Deflections (%)
	Load (N)	Deflection (mm)	Stiffness (K) (Nm/rad)	Load (N)	Deflection (mm)		
Tension							
R3A	24.09 (9.42)	4.36	460 (11.45)	24.13	4.19	−0.17	4.01
R3B	28.01 (13.11)	4.00	599 (9.29)	28.14	4.30	−0.46	−7.53
R4A	17.70 (8.05)	6.10	376 (14.12)	17.79	6.10	−0.49	0.00
R4B	27.07 (9.20)	3.24	589 (17.00)	27.08	3.24	−0.06	0.00
T3A	5.15 (5.15)	3.61	451 (13.63)	5.15	3.60	0.00	0.28
T3B	4.34 (5.51)	3.03	231 (7.42)	3.8	3.11	12.34	−2.64
T4A	9.55 (3.65)	4.00	306 (24.9)	8.96	4.74	6.13	−18.50
T4B	3.41 (11.46)	3.36	319 (12.65)	2.28	3.72	33.15	−10.71
RF	72.35 (10.31)	7.23	1975 (10.43)	73.99	6.99	−2.27	3.32
Compression							
R3A	7.48 (3.37)	3.96	470 (4.61)	7.54	3.69	−0.76	6.82
R3B	9.36 (0.60)	6.27	266 (10.07)	8.308	6.56	11.24	−4.63
R4A	5.14 (4.85)	2.06	313 (4.43)	5.45	2.10	−6.13	−1.94
R4B	9.27 (3.35)	5.90	286 (5.09)	8.72	4.69	5.88	20.51
T3A	0.00	0.00	0	1.97	0.10	NA	NA
T3B	2.54 (4.45)	0.36	298 (17.35)	2.55	0.45	−0.39	−25.00
T4A	0.00	0.00	0	3.8	5.03	NA	NA
T4B	0.00	0.00	0	1.49	2.83	NA	NA
RF	31.74 (8.71)	11.66	722 (10.84)	28.57	8.74	9.98	25.04

Values in parentheses are the coefficients of variations (COV), NA: Not applicable.

## Data Availability

The raw/processed data required to reproduce these findings cannot be shared at this time as the data also forms part of an ongoing study.

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
