# Peer review of "Analyses of L-Type Corner Joints Connected with Auxetic Dowels for Case Furniture"

_materials, 2023, doi:10.3390/ma16134547_

Round 1
Reviewer 1 Report
Dear authors
The paper was well written and presented. The experiment was appropariately design. However, there were some critical issues on the FEM and results. See followings.
1)It is recommended that the title should declare the subject clearly, "L-type case (panel) furnitrue corner";
2) In the Introduction section, it is recommended that review the FEM used in furnitrue structrue design, since you used it in your study;
3) In Materials and method section, the assemble process, or cut view of the joint should be presented in detail, which make it more readable;
4) The source of equation (1) should be cited;
5) In Fig. 10, for compression test, were the supports of the sample fixed? The freedom in transvese direction should not be fixed.;
6) In 2.5 section, the modelling methed is too simplified, especilly for auxetic dowels, inclusions only modelled in the center of the dowel, this is seriously different from the real Auxetic dowel. The expansion of auxetic dowel when subjected to tension is what you want to applied to increase the strength of joint, so it can not be ignored.
7) In results section, the maximum failure loads of samples connected by auxetic dowels were much lower than those control sample. Meanwhile, the maximum load of auxetic dowel joints do not satisfy the requirment of furnitrue.
8) The potential of auxetic dowel used in furnitrue is a doubted.
Author Response
Responses to Reviewers
We are grateful to the Editor and all the Referees for their time spent on our manuscript. The detailed responses are below. All corrections were added to the text in red and all additional explanations in blue.
Reviewer #1
Dear authors
The paper was well written and presented. The experiment was appropariately design. However, there were some critical issues on the FEM and results. See followings.
Comment 1: It is recommended that the title should declare the subject clearly, "L-type case (panel) furnitrue corner";
Answer 1: Based on the reviewer suggestion, the title has changed as “Analyses of L-type Corner Joints Connected with Auxetic Dowels for Case Furniture”.
Comment 2: In the Introduction section, it is recommended that review the FEM used in furnitrue structrue design, since you used it in your study;
Answer 2: Based on the reviewer suggestion, some pioneering studies were added to the paper related to the using FEM in furniture structures.
In a study, strength of a wooden chair was determined under different kinds of loads, then modelled the same chair and determined the stresses at some points under the same loads using FEM. It was reported that the analysis data and experimental data are very similar [20]. Based on a study regarding the using FEM in structural analyses of furniture; it is difficult to analyze the stresses that occur in chair frames, but FEM can be used to solve this problem. In this research, wooden chair frame was modeled and it was demonstrated how to analyze and design using FEM, and compared the analyses results to the real test results. In conclusion, it was stated that chair frames can be analyzed using computer aided structural analysis methods [21]. The difficulties of using FEM in wood materials and compared the performance tests with finite element software was investigated [22]. Experimental tests and FEM analyses were performed for different types of sofa frames constructed of wood and wood based materials. According to results, FEM results were given reasonable estimates about the strength properties of sofa frames. As a result, it was also emphasized that the joints are the critical points in furniture and that more durable joints can be made using materials with high bending strength [23].
- Gustafsson, S. I. 1996. Finite Element Modelling Versus Reality for Birch Chairs. Holz als Roh-und Werkstoff 54 (5): 355-359.
- Gustafsson, S. I. 1997. Optimising Ash Wood Chairs. Wood Sci Technol 31 (4): 291-301.
- Koç, K.H.; Kizilkaya, K.; Erdinler, E. S.; Korkut, D. S. 2011. The Use of Finite Element Method In The Furniture Industry. African Journal of Business Management Vol. 5(3), pp. 855-865, 4 February.
- Kasal, A.; Erdil, Y.Z.; Birgül, R. 2006. Determination of the strength performance of chair frames constructed of solid wood and wood composites. For. Prod. J. 56 (7-8): 55–60.
Comment 3: In Materials and method section, the assemble process, or cut view of the joint should be presented in detail, which make it more readable;
Answer 3: Authors think that the assembling process of the joint have already been explained in the manuscript and Figures 8 and 9 (currently Figures 9 and 10). According to the reviewer suggestion, the explanation was improved for easy understanding.
In the assembling of the specimens, holes 12 mm in diameter and 14.5 mm deep were first drilled into the face piece and 33 mm deep holes were drilled into the butt piece of the specimens to insert the face and butt muffs. The face and butt muffs were then fully inserted and bonded to the holes using Jowat® UniPUR 687.22 melamine based adhesive (Jowat Swiss AG, Buchrain, Switzerland). Prior to assembly, the dowels were manually inserted into the muffs placed on the face members. A Zwick 1445 universal testing machine (Zwick Roell AG, Ulm, Germany) with a loading rate of 10 mm/min was used to determine the dowel assembly forces (Fig. 10).
Comment 4: The source of equation (1) should be cited;
Answer 4: The source of equation (1) was cited with the studies below:
- KuÅŸkun, T.; Smardzewski, J.; Kasal, A. Experimental and Numerical Analysis of Mounting Force of Auxetic Dowels for Furniture Joints. Eng Struct 2021, 226.
- Kasal, A.; KuÅŸkun, T.; Smardzewski, J. Experimental and Numerical Study on Withdrawal Strength of Different Types of Auxetic Dowels for Furniture Joints. Materials 2020, 13, 1–21.
Comment 5: In Fig. 10, for compression test, were the supports of the sample fixed? The freedom in transvese direction should not be fixed.;
Answer 5: The reviewer is right. The freedom in transverse direction should be definitely free. This situation has already been explained in the manuscript. Furthermore, the Figure 10 (currently Figure 11) has been replaced with a more understandable picture.
For the tensile tests, the bottoms of each of the two members of the joint were placed on the pieces with rollers on the bottom and V-shaped grooves on the top, allowing the two parts of the joint to move outwardly as the corner joint was loaded (Fig. 11a).
Comment 6: In 2.5 section, the modelling methed is too simplified, especilly for auxetic dowels, inclusions only modelled in the center of the dowel, this is seriously different from the real Auxetic dowel. The expansion of auxetic dowel when subjected to tension is what you want to applied to increase the strength of joint, so it can not be ignored.
Answer 6: Yes, in the 2.5 section, the numerical model was simplified, especially for auxetic dowels. This simplification was explained and supported by an analysis of convergence (Fig. 15). In fact; the inclusions were modelled only in the place where the dowel connects two elements. Of course, the expansion of the auxetic dowel when subjected to the tension was what the authors wanted to apply to increase the strength of the joint. In Abaqus, dowels were partitioned into three parts: a solid shorter part, an inclusions part and a solid longer part. The mechanical properties of PA12, as in Table 2, were used for the inclusions part. In the case of the solid parts, mechanical properties of PA12, as in Table 2, were used, but negative Poisson's ratios were applied from Table 1 for each dowel type. So the auxetic phenomena on the whole length of the dowel were modelled correctly. The text uses an additional explanation resulting from the presented description.
In Abaqus, dowels were partitioned into three parts: a solid shorter part, an inclusions part and a solid longer part. The mechanical properties of PA12, as in Table 2, were used for the inclusions part. In the case of the solid parts, mechanical properties of PA12, as in Table 2, were used, but negative Poisson's ratios were applied from Table 1 for each dowel type.
Comment 7: In results section, the maximum failure loads of samples connected by auxetic dowels were much lower than those control sample. Meanwhile, the maximum load of auxetic dowel joints do not satisfy the requirment of furnitrue.
Answer 7: The number of studies on the possibilities of using auxetic materials in the furniture industry is very limited. The number of these studies should be increased, and accordingly auxetic materials should be introduced to the furniture industry instead of conventional materials because of the technical and economic advantages. This study is one of the first studies conducted for this purpose. As the reviewer said; according to the experimental results, the corner joints connected with auxetic dowels gave lower strength values than the corner joints connected with reference dowels. Of course, at the start of the study, designed and manufactured auxetic dowels were expected to yield higher results. However, the expected results could not be obtained with these dowels, in other words, the tested hypothesis of the study was rejected as a result of the experiments. Naturally, in every scientific study, a hypothesis is created, then tested, and either accepted or rejected as a result of experiment. In this study, the created hypothesis was rejected. However, the results of this study led to future studies to increase the strength of auxetic dowels and to investigate the new materials and alternative production technologies instead of 3D printing for dowels. These are mentioned in the conclusion section of the study.
Comment 8: The potential of auxetic dowel used in furnitrue is a doubted.
Answer 8: In general, it would not be correct to say that “the potential use of auxetic dowels in furniture joints is doubted” since the auxetic dowels tested in this study do not have sufficient strength. According to the feedback obtained from the results of this study, it will be possible to obtain stronger furniture joints by using re-designed auxetic dowels, new materials, and new production techniques instead of 3D printing.

Reviewer 2 Report
This research is interesting and there several suggestions as follows:
1. Please give more information on this joint connection application examples. Add one picture?
2. The test loading boundary conditions are shown in Figure.10. It is not clear on the base bottom displacement boundary. Is the bottom of the specimens fixed or allowed to slide?
3. In section “4. Conclusions”, it will be better to make a list but not one total paragraph.
Some sentences are not quite clear, please check.
Author Response
Responses to Reviewers
We are grateful to the Editor and all the Referees for their time spent on our manuscript. The detailed responses are below. All corrections were added to the text in red and all additional explanations in blue.
Reviewer #2
This research is interesting and there several suggestions as follows:
Comment 1: Please give more information on this joint connection application examples. Add one picture?
Answer 1: Authors think that the assembling process of the joint have already been explained in the manuscript and Figures 8 and 9 (currently Figures 9 and 10). According to the reviewer suggestion, the explanation was improved for easy understanding.
In the assembling of the specimens, holes 12 mm in diameter and 14.5 mm deep were first drilled into the face piece and 33 mm deep holes were drilled into the butt piece of the specimens to insert the face and butt muffs. The face and butt muffs were then fully inserted and bonded to the holes using Jowat® UniPUR 687.22 melamine based adhesive (Jowat Swiss AG, Buchrain, Switzerland). Prior to assembly, the dowels were manually inserted into the muffs placed on the face members. A Zwick 1445 universal testing machine (Zwick Roell AG, Ulm, Germany) with a loading rate of 10 mm/min was used to determine the dowel assembly forces (Fig. 10).
Comment 2: The test loading boundary conditions are shown in Figure.10. It is not clear on the base bottom displacement boundary. Is the bottom of the specimens fixed or allowed to slide?
Answer 2: The reviewer is right. The freedom in transverse direction should be definitely free. This situation has already been explained in the manuscript. Furthermore, the Figure 10 (currently Figure 11) has been replaced with a more understandable picture.
For the tensile tests, the bottoms of each of the two members of the joint were placed on the pieces with rollers on the bottom and V-shaped grooves on the top, allowing the two parts of the joint to move outwardly as the corner joint was loaded (Fig. 11a).
Comment 3: In section “4. Conclusions”, it will be better to make a list but not one total paragraph.
Answer 3: According to the rules of the journal, it is not appropriate to give the conclusions as a list. Therefore, this correction could not be performed.

Reviewer 3 Report
I think the language and structure of the paper must be deeply revised. The abstract is not clear: it should be more concise and clear. In general, the entire paper should include a section where the entire process is summarized (for example using bullet point).
The entire paper is too long and must be much more short and concise.
I really have difficulties to imagine how the results of this paper could be applied in real applications related with furniture. I understand that the proposed components have better mechanical properties but what about their price? Considering that generally furniture does not have structural safety issues why someone should replace the actual technology with another one much more complicated?
The first sentence of the introduction “Since the past, furniture has an important place in the possessions because it meets both the physical and psychological needs of the person.” Should not be found in scientific paper.
Tab.1 how did you determine the properties reported in the table? In particular, how did you determine the friction coefficient and the poisson ratio?
Fig.14. please comment more the figure because in table 5 and figure 21 the comparison between FE results and experiments seems good but in fig. 14 is really bad.
I see a great amount of work in this paper but unfortunately they are shown in a really bad way.
it must be deeply revised
Author Response
Responses to Reviewers
We are grateful to the Editor and all the Referees for their time spent on our manuscript. The detailed responses are below. All corrections were added to the text in red and all additional explanations in blue.
Reviewer #3
Comment 1: I think the language and structure of the paper must be deeply revised. The abstract is not clear: it should be more concise and clear. In general, the entire paper should include a section where the entire process is summarized (for example using bullet point).
Answer 1: The language and structure of the study were extensively reviewed and corrected. The abstract has been completely re-organized.
Tests were carried out to develop and manufacture various types of auxetic dowels using 3D printing technology. These dowels were then used to connect L-type corner joint specimens for case furniture, and their strength and stiffness were analyzed through experimental, theoretical, and numerical means. In the scope of the study, 8 different types of auxetic dowels including 2 inclusion types, 2 inclusion sizes and 2 dowel hole diameter; and a reference non-auxetic dowel were designed. Accordingly, a total of 180 specimens that include 10 replications for each group were tested; 90 were tested under tension and the remaining 90 under compression. The results demonstrated that the assembly force required for the corner joints connected with auxetic dow-els was significantly lower compared to non-auxetic dowels. Furthermore, the numerical and theoretical analyses yielded similar outcomes in this study. Both analyses revealed that the dowels used to connect the corner joints experienced substantial stresses during mounting and bending, ultimately leading to their failure. Upon concluding the test results, it was observed that the corner joints connected with dowels featuring rectangular inclusions exhibited superior performance when compared to those with triangular inclusions. In light of these findings, it can be concluded that further enhancements are necessary for auxetic dowels with rectangular inclu-sions before they can be utilized as alternative fasteners for traditional dowels.
Comment 2: The entire paper is too long and must be much more short and concise.
Answer 2: The work is very extensive. The content of the study includes both experimental part, numerical analysis and theoretical calculations. All these sections have been tried to be given with an appropriate methodology. The entire article has been reviewed and tried to be shortened as much as possible. Furthermore, in order to make the article more understandable, a flowchart and an explanation containing the methodology of the study was created and added to the beginning of the Materials and Methods section (Figure 1).
In the methodology of the study; in the first stage, the design, analysis and production of the auxetic dowels were carried out. In the second stage, the elastic properties of the produced dowels and the elastic and mechanical properties of the material (PA12) used in the production of the dowels were determined. In the next step, the strength and stiffness of the corner joints connected with the designed and manufactured dowels were analyzed both experimentally, numerically and theoretically. In the last stage, the results obtained from the experiments, numerical analyzes and theoretical calculations were compared and interpreted.
Comment 3: I really have difficulties to imagine how the results of this paper could be applied in real applications related with furniture. I understand that the proposed components have better mechanical properties but what about their price? Considering that generally furniture does not have structural safety issues why someone should replace the actual technology with another one much more complicated?
Answer 3: The aim of engineering design is to manufacture products in the ideal intersection of technical and economic considerations. Sometimes weak strength products are strengthened, while sometimes unnecessary excessive strength products are reduced to a sufficient strength level, resulting in economic gain. The auxetic dowels within the scope of the study is designed to be used in corner joints of case-type furniture. This fastener has advantages over the other fasteners commonly used in the furniture industry. These are,
- Significantly lower cost
- Ease of assembly
- Reducing production operations and diversity
- There is no need for any tools for assembly, it can be easily assembled by hand.
Accordingly, the use of auxetic dowels as an alternative to traditional fasteners will provide significant technical and economic advantages to consumers and manufacturers.
Comment 4: The first sentence of the introduction “Since the past, furniture has an important place in the possessions because it meets both the physical and psychological needs of the person.” Should not be found in scientific paper.
Answer 4: The first sentence of the introduction was taken out of the study.
Comment 5: Tab.1 how did you determine the properties reported in the table? In particular, how did you determine the friction coefficient and the poisson ratio?
Answer 5: An explanation was improved into the related part of the manuscript.
The exact diameter of the dowel and the inner diameter of the muff were measured individually with a digital caliper. Uniaxial compression tests were performed on all dowel groups to calculate the coefficient of friction and Poisson's ratio of the dowels. In order to obtain the Poisson’s ratio of dowels; a reference ruler was placed behind the dowels. Two pictures of the dowels were taken one before loading and the other at the time of 2 mm deformation in vertical (Y) direction. Then, dowel strains in vertical and horizontal directions were analyzed using the National Instruments IMAQ Vision Builder 6.1 software (National Instruments, Austin, TX, USA). Poisson’s ratios were calculated by applying the edge detection method in the digital image analysis. Since the methodology for determining the coefficients of friction has been described in detail in [6,7], only the final results of the calculations in Table 1 are presented in this part of the paper. They were used for further numerical calculations.
Comment 6: Fig.14. please comment more the figure because in table 5 and figure 21 the comparison between FE results and experiments seems good but in fig. 14 is really bad.
Answer 6: The authors would like to point out that Figure 14 (currently 15) presents only the results of the convergence assessment of a randomly selected joint's model R4B, based on which it was decided to choose the size of the finite element mesh for all other numerical models (Of course, the relevant explanations are given in the text of the manuscript, above, figure 15). According to the results obtained from these preliminary analyses, the most appropriate mesh size was accepted as 4 mm, and this size was used in the FEM analysis. Thanks to them, the results of the numerical calculations and the results obtained from the real experiments were relatively consistent. As mentioned by the Reviewer, the consistency between the results is also evident in Table 5 but in Figure 15 only for R4B.
Comment 7: I see a great amount of work in this paper but unfortunately they are shown in a really bad way.
Answer 7: The structure of the article was reorganized as much as possible based on the reviewer’s recommendations. Many thanks to the referee for this review. In this way, the article became easy to understand and its scientific quality increased. Authors think that; the promising potential of auxetic materials should not be ignored for the furniture industry as well as for every sector. The research topic is unique, and the outputs to be obtained as a result of this study will not only bring a new approach to the furniture industry, but will also lead to the production of new qualified scientific publications in the future.

Round 2
Reviewer 1 Report
Dear authors
Although some issues were not well adressed, i think this paper can be accepted for publication. Hope to see your future work on this topic.
Reviewer
Author Response
Thank you for all the suggestions that appreciated our effort and helped to improve our work.